# Nanohybrid Membrane Synthesis with Phosphorene Nanoparticles: A Study of the Addition, Stability and Toxicity

**DOI:** 10.3390/polym12071555

**Published:** 2020-07-14

**Authors:** Joyner Eke, Philip Alexander Mills, Jacob Ryan Page, Garrison P. Wright, Olga V. Tsyusko, Isabel C. Escobar

**Affiliations:** 1Center of Membrane Sciences, Department of Chemical and Materials Engineering, University of Kentucky, 177 FPAT, Lexington, KY 40503, USA; joyner.eke@uky.edu (J.E.); philipamills2001@gmail.com (P.A.M.); jacob-page@uky.edu (J.R.P.); 2Department of Plant and Soil Sciences, University of Kentucky, 1100 S. Limestone St., Lexington, KY 40506, USA; garrison.wright@uky.edu (G.P.W.); olga.tsyusko@uky.edu (O.V.T.)

**Keywords:** phosphorene, fouling reduction, two-dimensional materials, leaching, membrane modification, nanoparticles, reactive membranes, toxicity study

## Abstract

Phosphorene is a promising candidate as a membrane material additive because of its inherent photocatalytic properties and electrical conductance which can help reduce fouling and improve membrane properties. The main objective of this study was to characterize structural and morphologic changes arising from the addition of phosphorene to polymeric membranes. Here, phosphorene was physically incorporated into a blend of polysulfone (PSf) and sulfonated poly ether ether ketone (SPEEK) doping solution. Protein and dye rejection studies were carried out to determine the permeability and selectivity of the membranes. Since loss of material additives during filtration processes is a challenge, the stability of phosphorene nanoparticles in different environments was also examined. Furthermore, given that phosphorene is a new material, toxicity studies with a model nematode, *Caenorhabditis elegans*, were carried out to provide insight into the biocompatibility and safety of phosphorene. Results showed that membranes modified with phosphorene displayed a higher protein rejection, but lower flux values. Phosphorene also led to a 70% reduction in dye fouling after filtration. Additionally, data showed that phosphorene loss was negligible within the membrane matrix irrespective of the pH environment. Phosphorene caused toxicity to nematodes in a free form, while no toxicity was observed for membrane permeates.

## 1. Introduction

Membranes play a crucial role in the purification of water and wastewater [1]. Within the broad range of membrane materials, polymeric membranes are attractive because they exhibit high chemical and mechanical resistance and offer a wide range of pore sizes; however, polymeric membranes are plagued by fouling, which is a problem that has hindered fast adaptation of membranes in relevant fields [2]. Fouling is the buildup of unwanted materials on the membrane surface and within the pore structure. Fouling materials are grouped under three generic headings, namely organic foulants (proteins, humic and other organic compounds), inorganic foulants (mineral salts, crystallized salts, oxides and hydroxides and colloidal particles) and biologic foulants (biofilm formation by microorganisms) [3]. Fouling inhibits membrane performance as measured by permeability and selectivity, increases membrane maintenance costs, and ultimately shortens the lifespan of the membrane [4]. Membranes can be functionalized with reactive nanomaterials to improve their fouling resistance properties [5]. Dynamic/reactive membranes can mitigate fouling by the generation of reactive oxygen species, which oxidize foulants present on the membranes [6] thus leading to a self-cleaning phenomenon.

Two-dimensional materials are being increasingly researched as membrane additives since they create ultrathin separation layers within the membrane that are highly selective for molecules and ions [7]. Two-dimensional nanomaterials are materials that can be isolated as freestanding one-atom-thick sheets [8]. One of such two-dimensional materials is phosphorene, which was discovered in 2014 [9], and was found to exhibit improved optical properties; it displays optical absorption peaks at 1.2 eV and absorbance spectrum across both the IR (infrared) and visible light spectra [10]. These optical properties can be explored in photocatalysis, hence, it can be considered to be a potential metal-free photocatalyst [11]. In the study done by Yang et al. by applying density functional theory calculations [12], phosphorene was shown to display photocatalytic hydrogen production properties. Phosphorene is a single layer, two-dimensional layered material, exfoliated form of black phosphorus (BP). Unique properties of phosphorene include its highly anisotropic electric conductance and its strong interaction with light. Phosphorene distinguishes itself from other 2D-layered materials by its intrinsic structural anisotropic features [13]. Unlike graphene, phosphorene combines a high carrier mobility with a fundamental band gap [14] which imparts an intrinsic fine-tuning ability [15], thereby providing numerous opportunities for research. The main issue with phosphorene is the fast degradation under ambient conditions as a result of the generation of reactive oxygen species. Research however has shown that incorporating phosphorene into polymers preserves the structure and properties of phosphorene [16,17]

Specifically relevant to the field of liquid separations using membranes, the band gap of phosphorene provides it with electronic [18] and photocatalytic [19] properties, which could be explored in making responsive membranes that could simultaneously remove and destroy organic compounds. Phosphorene has recently been used as a catalyst for arsenic removal [20] and as a photocatalyst for dye degradation [6]. However, a key issue with phosphorene is instability when exposed to air, which causes it to degrade into phosphorus oxides that may affect its chemical and physical properties [21]. Several studies have focused on addressing this issue, such as Ryder et al. produced phosphorene nanoparticles that were stable in ambient conditions for three weeks by chemically modifying exfoliated black phosphorus with an aryl diazonium molecule which formed covalent phosphorus–carbon bonds and increased their stability [22]. Recently, Qiu et al. synthesized phosphorene, which exhibited stability when exposed to ambient conditions for four months by crosslinking black phosphorus with polyphosphazene [23].

Integrating nanoparticles within polymeric membrane matrices could lead to increases in their selectivity, thermal stability, permeability as well as altering their water affinity characteristics [24]. Nanoparticles can be prepared using physical processes that utilize a top-down technique (breaking down the bulk material into nanoparticles) or chemical processes which utilize bottom-up techniques (typically employ chemical reactions to assemble atoms together) [25]. Several techniques for incorporating these nanoparticles into membranes include layer-by-layer assembly, chemical grafting, self-assembly and physical deposition, among others [26]. Common problems of aggregation and/or leaching of the nanoparticles may occur irrespective of the technique used for incorporating them into the membranes. Nanoparticle agglomeration occurs as a result of the very attractive forces between the nanoparticles, such as van der Waals and electrostatic forces [27]. While converting bulk crystalline solids into spherical nanoparticles, there is an energy loss associated with the deformation of the particles. At the point of contact between two nanoparticles, an adhesive grain boundary is formed that is thermodynamically stable. The energy at the free surface of these nanoparticles is two times higher than the energy at the grain boundary. As a result, whenever two nanoparticles come in contact, there is always an energy gain hence agglomeration occurs [28]. To prevent agglomeration, an opposite repulsive force is required [25]. When phosphorene is made by exfoliating bulk crystalline black phosphorus, agglomeration may occur.

Leaching of nanoparticles from polymer media is a common phenomenon because, stabilizing nanoparticles in aquatic environments is intricate as a result of the Brownian diffusion that largely controls particle movements [29]. Other interactions that govern nanoparticle stability includes steric, hydration and magnetic forces. Coagulation of nanoparticles in a solvent media can be prevented by stabilizing with a polymer because they can induce steric stabilization in the particles [30]. Although with a weak solvent, the van der Waal interactions can dominate and cause the polymer layer to collapse leading to coagulation [31]. Among factors governing how the polymer interacts with the nanoparticle are the technique used in coating the nanoparticle with the polymer (adsorption vs grafting), the level of coverage and nature of the polymer [32].

Phosphorene is a metal-free photocatalyst and provides advantages over application of toxic metal-based photocatalysts such as oxides, sulfides and nitrides of titanium, tungsten, cadmium and transition-metal dichalcogenides. Several studies have examined in vitro and in vivo toxicity of phosphorene and demonstrated that it can cause toxicity [33,34]. The observed cytotoxicity from phosphorene in one of the studies was lower than that of graphene [35]. Mice exposed to black phosphorus quantum dots after showing signs of oxidative stress were able to recover from the exposure [36]. There is evidence that phosphorene nanosheets can penetrate cell membranes and interact with phospholipid layers, and the degree of these interactions as well as resulting toxicity are determined by the size and concentrations of phosphorene and the cell types [37]. It is still unknown, however, whether phosphorene embedded into polymeric membranes will be released at the concentrations that could cause toxicity. In this study, we utilized a powerful model organism, a nematode *Caenorhabditis elegans* to test for in vivo toxicity of phosphorene. Due to their short generation time, ease of maintenance and prolific reproduction *C. elegans* has been extensively used as a model organism for a toxicity testing of various contaminants including nanomaterials [38,39]. In addition, its genome is fully sequenced, annotated, and functional genomic tools are readily available for examining toxicity mechanisms.

As more researchers turn to two dimensional materials for membrane modifications, the need for a 2D material that inherently allows fine tuning towards membrane enhancement is pertinent. Graphene has no band gap and other 2D transitional metal dichalcogenides (TMDs) possess band gaps only as monolayers [40]. Phosphorene has direct band gaps in all its three forms, bulk, monolayer and few layers [40]. Phosphorene has also been studied for its electrocatalytic properties, which research shows outperforms ruthenium (iv) oxide and Co_3_O_4_/N-graphene [41]. Currently, while a large bulk of experimental research efforts has focused towards producing air stable phosphorene [40,41,42,43], there is limited information on the incorporation of phosphorene in membranes as well as a thorough understanding of its physicochemical properties when utilized as a membrane additive In a previous study [6], the photocatalytic properties of phosphorene-based membranes were examined; on the other hand, in this study, the effects of phosphorene on the morphologic structure of the polymeric blend along with the evolution of the modifications were investigated. Furthermore, we discuss its stability under several pH environments as well as study biologic effects of phosphorene-based membrane permeates on a nematode.

The overarching goal is to develop stable and non-toxic phosphorene polymeric membranes. To achieve this goal the prepared phosphorene membranes were thoroughly characterized with respect to their structural and morphologic characteristics, permeability and selectivity, as well as toxicity. Our specific objectives were to (1) incorporate phosphorene into a polymer blend of polysulfone (PSf) and sulfonated poly ether ether ketone (SPEEK) to cast ultrafiltration membranes; (2) examine stability of phosphorene in acidic, basic and neutral environments; (3) determine the level of adhesion of phosphorene to the membranes via leaching experiments with a closed cross flow filtration; (4) examine toxicity of free phosphorene and permeates of phosphorene membranes to *C. elegans*.

## 2. Materials and Methods

### 2.1. Materials

To produce few-layered phosphorene, bulk black phosphorus was purchased from Smart Elements, Vienna, Austria. Polysulfone (PSf), poly ether ether ketone (PEEK), *N*-methyl pyrrolidone (NMP), used to prepare the dope solution for ultrafiltration membranes, were purchased from VWR, Radnor, PA, USA. Methylene blue was purchased from VWR, Radnor, PA, USA. Sodium hydroxide (NaOH), bovine serum albumin (BSA), sodium chloride (NaCl), concentrated sulfuric acid, phenolphthalein indicator and citric acid were also purchased from VWR, Radnor, PA, USA. The cross flow cell was designed in the laboratory. The ultrasonicator model P70H was purchased from Elmasonic P, Singen, Germany. A dead-end cell, Amicon stirred-cell 8010–50 mL, was purchased from EMD Millipore, Burlington, MA, USA. Total organic carbon analyzer TOC-5000A was purchased from Thermo Scientific, Waltham, MA, USA.

### 2.2. Sulfonation of PEEK and Determination of Degree of Sulfonation

The recipe for making the SPEEK polymer dope was previously reported in the literature, so it is briefly discussed here [6]. To synthesize sulfonated poly ether ether ketone, PEEK pellets were dried in the oven at a temperature of 60 °C overnight and then dissolved in a 98% concentrated sulfuric acid solution for three days at room temperature. After dissolution, it was gradually added into an ice water bath under mechanical agitation to precipitate SPEEK (sulfonated poly ether ether ketone) pellets. SPEEK was thoroughly washed in deionized water until a pH of 7 was attained. Then it was dried in the oven at 60 °C and stored for use.

The degree of sulfonation (DS) is the content of hydrogen sulfite present after all possible substitution for hydrogen sulfite has occurred on all points in the substitution site [44]. For SPEEK, sulfonation usually happens on the phenyl ring located between the two ether groups of the PEEK repeat unit [45]. SPEEK with a high degree of sulfonation (DS) has a relatively low chemical stability [46]. Hence, it is necessary to calculate DS. To quantitatively determine the DS, the ^1^H NMR spectra of SPEEK in a deuterated solvent, dimethyl sulfoxide (DMSO-*d*_6_) was carried out. At a frequency of 400 MHz, a Bruker (Billerica, MA, USA) Avance NEO spectrometer equipped with a Smart Probe was utilized for this experiment. A total of 5 wt % of SPEEK was dissolved in DMSO-*d*_6_. The internal standard used was DMSO at 2.5 mg/L. The assignment of the peak signals was from literature [47]. The presence of –SO_3_H group after substitution results in a down field shift of the nearest neighboring proton (*H*_10_) as seen in Figure 1. Using Equation (1), the DS was estimated from the ratio between the peak area of *H*_10_ and the total integrated peak area of all the remaining aromatic protons (*H_x_*, where *x* = 1, 2, 3, 4, 5, 6, 7, 8, 9, 11) [44].
(1)$$\frac{Peak\;area\;of\;distinct\;signal\;\left(H_{10}\right)}{\sum Peak\;area\;of\;remaining\;aromatic\;signals}=\frac{y}{12-2y}=\frac{Area\;of\;H_{10}}{\sum Area\;of\;H_{X}}\;(0\;\leq \;y\;\leq \;1)$$

### 2.3. Fourier-Transform Infrared Spectroscopy (FTIR)

FTIR is commonly used in the study, identification, degradation and characterization of polymeric structures [48]. Molecules can absorb light in the infrared region that usually translates into changes in the vibrational frequency [49]. Apart from diatomic elemental gases, all compounds exhibit infrared spectra and can be analyzed qualitatively by their distinctive infrared absorption [49]. Functional groups possess characteristic infrared absorption bands that are synonymous to the stretching, contracting and bending vibrations of the functional groups. These vibrations are expected within specific regions on the spectra and are influenced by the kind of chemical bonds present in the functional group, as well as the atoms which make up the group [50]. The infrared region of the spectra is divided into three basic regions which are the near-IR, mid-IR and far-IR. The mid-IR which falls under wavelength numbers spanning from 400 up to 4000 cm^−1^ is where most chemical molecules absorb frequencies and exhibit vibrations [51].

### 2.4. Exfoliation of Bulk Black Phosphorus

To produce few-layer phosphorene, the method described by Guo et al. and Eke et al. was utilized [6,52]. Briefly, equal volumes of NMP and NaOH were mixed and degassed on an ultrasonicator (P70H, Elma Elmasonic P, Singen, Germany) for 5 min. Three hundred milligrams of bulk black phosphorus was suspended in this mixture and sonicated for 5 h at a frequency of 37 KHz and a power of 80%. The temperature was kept constant throughout the experiment at 30 °C. The solution was centrifuged at 4000 rpm for 23 min. The supernatant was used for the experiment.

### 2.5. Transmission Electron Microscopy (TEM) and HAADF–STEM

Exfoliated phosphorene samples were prepared and added dropwise onto a carbon film on a copper grid (Lacey carbon film, 300 Mesh Cu, TED Pella, Inc., Redding, CA, USA). The lacey carbon film was then left in a hood overnight to completely dry the solvent. High-resolution transmission electron microscopy (HR-TEM) was performed on a FEI Talos F200X, Waltham, MA, USA, instrument operated at an accelerating voltage of 200 kV and with a point-to-point resolution of 0.1 nm. The TEM images were obtained at typical magnifications of 100 K to 1.05 M Velox digital micrograph software was used to analyze the samples and Image J was used to estimate nanoparticle size. Scanning transmission electron microscopy (STEM) was performed on the same instrument (FEI Talos F200X, using a high angle annular dark field Detector (HAADF) at 200 kV. HAADF–STEM image intensity is reported to be proportional to square of the atomic number, so heavy atoms are observed brighter. The phosphorene nanoparticle composition and element distribution were determined via FEI super energy-dispersive X-ray spectroscopy (EDX) system.

### 2.6. Optical Profilometer

The surface morphology of membranes influence the fouling pattern, membrane permeability as well as the solute rejection of the membrane [53]. Studies have shown that smoother surfaces tend to exhibit lower rejection and higher flux values, whereas, rougher surfaces exhibit higher rejection and lower flux values [54]. Atomic force microscopy (AFM) is the most common technique used for characterization of membrane surface roughness because of ease of use and functionality in different environments but a major drawback is the limitation on scan surface area [55]. Given that surface roughness is a function of scan size, a small scan area may be misrepresentative of the true overall surface roughness [56]. Optical interferometry on the other hand, provides roughness information over a larger scan size and thus more accurate information can be deduced on the membrane surface roughness [55]. Optical profilometers are used to evaluate height variations on surfaces hence information on the surface roughness of the surface can be obtained. They are interference microscopes that utilize the wave properties of light to compare the optical path difference between a test surface and a reference surface. Surfaces can be characterized quickly and precisely to determine surface roughness, critical dimensions and any additional topographic features. Measurements made are usually nondestructive and do not require sample preparation. The Zygo New View 7000 optical profilometer (Zygo Corporation, Middlefield, CT, USA) was used to characterize the surface of the phosphorene modified membrane as well as the unmodified membrane. The scan length was 65 µm bipolar (20 s) and the magnification of the objective lens was 50×. Two dimensional and three-dimensional images were obtained for analysis.

### 2.7. Electrokinetic Potential Measurement

The zeta potential provides information on the surface charge of the membrane surface. An Anton Paar SurPASS electrokinetic analyzer (Anton Paar, SurPASS, Ashland, VA, USA) was used to determine the zeta potential of the membrane. The electro kinetic potential, commonly referred to as the zeta potential is the potential at the shear plane of a moving colloid particle under an electric field. It describes the potential difference present between the electric double layer of the particles in motion and the stationary layer of dispersant surrounding these particles at the slipping plane [57]. Certain factors control the observed value for zeta potential which include pH, positive in acidic environments and negative in basic environments, ionic strength, the higher the ionic strength the lower the zeta potential, concentration, dilute solutions have a higher zeta potential [57,58,59].

### 2.8. Contact Angle Measurement

The water interaction parameter of a membrane surface plays a key role in water permeability and fouling [60]. For a drop of liquid on a horizontal flat surface, the contact angle is the angle between the juncture of the solid–liquid boundary and the vapor–liquid boundary [61]. When the contact angle of a surface is less than 90° it implies that the surface is a high level of wettability and hydrophilic, while surfaces with contact angles greater than 90° indicates a low amount of wettability and hydrophobicity. For this study, a drop shape analyzer (Kruss DSA100, Matthews, NC, USA) was used to obtain contact angle measurements on all the membrane samples.

### 2.9. Leaching Studies

The stability of static phosphorene within the pores of the membrane was examined using a cross flow cell, the schematic is shown in Figure 2 in recycle mode. The feed solution, deionized water, was stirred at a rate of 200 rpm. The phosphorene-embedded membrane was left in continuous contact with the deionized water flowing throw the cell for fifteen days. Samples were taken daily from the beaker and tested for presence of phosphorus using an inductively coupled plasma atomic emission spectroscopy (ICP-OES). During the membrane formation process via phase inversion in a water bath, samples of the remnant water-solvent mixture were obtained and tested for phosphorene. Furthermore, the stability of the phosphorene membranes at various pH levels was tested at room temperature. For the stability studies, concentrations of phosphorene in the dope solutions were 650, 800 and 1000 mg/L. Phosphorene membranes with an area of 100 cm^2^ were left in 100 mL of citric acid at pH 4, sodium hydroxide at pH 13 and deionized water at pH 7, respectively for 72 h. and samples were collected and analyzed for phosphorus using the ICP-OES. The detection limit was 60 ppb.

### 2.10. Inductively Coupled Plasma-Optical Emission Spectrometry (ICP-OES) Study

A Varian Vista Pro CCD simultaneous ICP-OES was used to determine the concentration of phosphorus in samples. The power used was 1.2 kW, plasma flow rate of 15 L/min, auxiliary flowrate of 1.5 L/min, nebulizer flowrate of 0.9 L/min, the replicate read time was 35 s and the instrument stabilization delay was 20 s. Samples were acidified to a pH < 5.5. A 25-ppb analytical detection limit was established with phosphorus calibration standards prepared in 1% HNO_3_. Standard curve correlations maintained a correlation coefficient >0.995. Sample measurements were read in triplicate. Quality control measures included a diluent blank, standard control and yttrium internal standard measurements with each sample reading. The ICP-OES was also utilized to measure phosphorous concentrations in free phosphorene exposure solutions used in toxicity testing (described below) as well as in permeates generated by filtering media through phosphorene membrane.

### 2.11. Morphologic Characterization of Membranes Using X-ray Photoelectron Spectroscopy (XPS) and Scanning Electron Microscope (SEM)

XPS characterization of phosphorene membranes was performed using a Thermo Scientific K-Alpha XPS, Waltham, MA, USA, apparatus equipped with an Al K (1486.6 eV) source (pass energy of 20 eV). Phosphorus, carbon, oxygen, and sulfur peaks were fitted using Thermo Scientific™ Avantage Software, Waltham, MA, USA. The X-ray source had an emission current of 12 mA and the acceleration voltage was 10 kV. The spectra measurement was done at a 90° emission angle. The electron energy analyzer operates in fixed analyzer transmission (FAT) mode, with a constant pass energy of 50 eV for survey (wide) scans and 20 eV for high resolution scans. The overall resolution of this XPS was about 1.1 eV. Furthermore, a depth profile scan for phosphorus was done to confirm presence of phosphorus on the surface of the membrane and within the pores of the membrane.

For the SEM characterization, the membranes were first immersed and ruptured in liquid nitrogen to obtain a fractured surface with minimal deformation (stretching and tearing). The resulting fracture, cross-section surfaces were then imaged in a scanning electron microscope (SEM, Quanta FEG 250, FEI/Thermo Fisher Scientific, Waltham, MA, USA) without conductive coating.

### 2.12. Membrane Synthesis

Optimal materials for membranes should have a blend of high permeability and selectivity, excellent mechanical strength, great film-forming properties and chemical and thermal stability [62]. Finding a polymer with all these attributes may be difficult and hence it is much easier to use polymer blends that can combine together to achieve these characteristics [63]. To make the membrane used for this experiment, a blend of polymers was utilized. PSf has a high thermal and chemical stability, but poor solubility in solvents and hydrophobic, while sulfonated poly ether ether ketone is hydrophilic, but has poor permeability. A blend of the two polymers gives rise to a membrane with high permselectivity and a superior permeability for water [64]. The dope solution consisted of a (95/5%) ratio of PSf and SPEEK and 0.5 wt % of exfoliated phosphorene in NMP. During the phase inversion process, some loss of phosphorene may have occurred, but this was unnoticeable. The remnant coagulant bath solution was tested after casting for phosphorus which was below the detection limit of 50 ng/mL. A total of 0.5% *w*/*v* of phosphorene was used (5 mg/mL) during the fabrication of the membrane and since no loss was detected; therefore, the theoretical percentage of phosphorene in the membrane was 0.5% *w*/*v*. Since nanoparticles can change the morphologic structure of membranes by acting as pore formers, keeping the concentration to a low 0.5 wt % helped balance the tradeoff of their positive impact on their negative impacts on the membrane [65,66].The solubility parameter for the blend polymer mixture and solvent were very close, which further indicated better compatibility [67] because it leads to a smaller heat of mixing hence increasing the possibility of a negative Gibbs free energy favoring a stable solution mixing [68]. SPEEK has a solubility parameter of 26.1–26.4 MPa^1/2^ [69,70]. Polysulfone has a solubility parameter of 23.7 MPa^1/2^ and NMP of 23.1 MPa^1/2^ [71,72,73]. Using physical mixing between the blended membrane polymer dope and phosphorene, Van der Waals interactions were formed between the constituents, and hence, phosphorene nanoparticles were incorporated into the dope solution. The membranes were cast using a doctor’s blade via the non-solvent induced phase separation technique. Figure 3 highlights the major step involved for the fabrication technique. The membranes were stored in deionized water overnight to further eliminate residual solvents.

### 2.13. Flux Analysis

To study the flux performance of the membrane, a 50-mL dead-end filtration cell was used under continuous stirring in a batch mode. A Whatman filter paper (110 mm) was used as a support for the membranes during the experiment. The filtration was done under a constant pressure of 2.06 bar at room temperature. The time for 2 mL of solution to pass through membranes with an area of 13.4 cm^2^ was recorded, and the flux, J, was calculated using this Equation (2).
(2)$$J=\frac{V}{A\Delta t}$$
where V is the volume of solution through the membrane in L, and A is the active filtration area of the membrane cell in m^2^, and t is the permeation time. Precompaction using deionized water was done before the filtration of bovine serum albumin (BSA) feed solution. This was repeated 10 times and then followed by a reverse-flow filtration of deionized water to simulate cleaning to eliminate foulants and determine flux recovery of the membrane. Using Equation (3), the protein rejection of the membrane was calculated.
R = (1 − (C_p_/C_f_)) × 100%(3)
where C_p_ is the protein concentration in the permeate and C_f_ is the protein concentration in the feed.

In addition to BSA, 10 mg/L of an organic dye, methylene blue (MB), was also filtered through the membranes and exposed to visible light and ultraviolet light (Spectroline Model EA-160, Westbury, NY, USA) at a wavelength of 365 nm for 30 min and visible light and the membranes were examined under a fluorescent microscope (Zeiss 880 NLO, Thornwood, NY, USA). The fluorescent intensity was analyzed using ImageJ to evaluate the percentage coverage of methylene blue on the surface.

### 2.14. Toxicity Testing

For toxicity testing phosphorene was transferred from solvent into DI water for triple washing. The washing steps required centrifugation at 12,000 g for 20 min, removal of supernatant leaving phosphorene pellet intact on the bottom of the tube and replenishing with DI water. After third wash, the exposure medium was added. Two media used for the exposures were 50% K medium (31.68 mM KCl and 51.37 mM NaCl) and moderately hard reconstituted water (MHRW; KCl 4 mg/L, MgSO_4_ 60 mg/L, CaSO_4_ 60 mg/L, NaHCO_3_ 96 mg/L) [74]. The protocols for toxicity screening were modified from previously established *C. elegans* toxicity testing methods [75,76]. Wild-type N2 strain of *C. elegans* were obtained from Caenorhabditis Genetics Center (CGC). The nematodes were age-synchronized and the eggs placed on K-agar plates with *Escherichia coli* OP50 as a food source [77]. For mortality, the L3 stage nematodes were exposed to concentration range of Phosphorene from 0 to 60 mg/L in two media, 50% K medium The 24-well tissue culture plates were used for exposures with 1 mL of the solution Ml concentrations was conducted in two independent experiments. For reproduction, eggs were hatched on K-agar plates with *E. coli* OP50 bacterial lawn, and after 24 h the nematodes at F2 stage were placed into the exposure solutions for 24 h. In each treatment, the nematodes were exposed to four sublethal concentrations of Phosphorene in 50 K medium or MHRW. The exposures were conducted in the presence of bacterial food, *E. coli* OP50 at OD_600_ = 1 and 10 μL per mL of exposure solution. After exposures individual nematodes were placed on K-agar plates containing *E. coli* OP50 and allowed to reproduce. The adults were transferred to the new K-agar plates every 24 h up to 72 h. The offspring that remained on the plate were allowed to hatch and grow for 24 h and after that were stained with rose bengal (0.5 g/L) and heated at 55 °C for 50 min. The stained offspring were counted under microscope. Mortality testing were also conducted with 1, 3, and 5 order of permeates generated after filtering K medium or MHRW through phosphorene membrane.

## 3. Results and Discussions

### 3.1. Degree of Sulfonation and Membrane Fabrication

In previous studies, the fabrication of SPEEK membranes along with the incorporation of phosphorene has been discussed [6]. As previously stated, the degree of sulfonation (DS) is the content of hydrogen sulfite present after all possible substitution for hydrogen sulfite has occurred on all points in the substitution site [44], and SPEEK with a high DS has a relatively low mechanical stability [46]. As seen in Figure 4, the peak from H_10_ was a doublet (two close peaks) at 7.5 ppm. The peaks from other protons far away from the carbonyl group, H_1,2,7,8,9,11_ was noticed at 7.0–7.3 ppm and the remaining peaks at 7.8–8 ppm. From the ^1^H NMR, the degree of sulfonation was measured by presetting the integration value of the distinct signal to 1.00 and then obtaining the integration values of the remaining signals from the spectra. These numbers were inserted into Equation (1) and the value obtained was 0.77. This means that the chances for SPEEK leaching out of the blend polymer membrane of PSf and SPEEK was low at room temperature since for SPEEK to dissolve out of the blend, the DS has to be greater than 0.99 [78]. Hence, the SPEEK membranes were considered chemically stable at room temperature.

A degree of sulfonation of 0.77 verified that the membranes would not solubilize during filtration and further supported the recipe used here. Therefore, as previously stated, the base membrane dope solution consisted of a blended polymer prepared by dissolving PSf and SPEEK in a 95%:5% ratio, respectively, in NMP. Using physical mixing between the blended membrane polymer dope and phosphorene, Van der Waals interactions were formed between the constituents, and hence, phosphorene nanoparticles were incorporated into the dope solution.

### 3.2. Structural Membrane Polymer Evolution

To verify the blending of PSf with SPEEK and determine if phosphorene led to alterations in the base polymeric backbone of the membranes, FTIR was performed. Figure 5 shows the FTIR bands at 400–4000 cm^−1^ of both unmodified and phosphorene membranes. Polysulfone displays characteristic bands at 1487 and 1586 cm^−1^ [79] which are due to the stretching vibration of the C=C aromatic ring. Similar bands were observed for both membranes, showing distinctive bands 1487 and 1586 cm^−^^1^ that verify the presence of PSf Furthermore, the presence of SPEEK was verified by characteristic broad bands at 3450 cm^−1^ from the hydroxyl group vibrations of SO_3_H, 1230 cm^−1^ assigned to the vibrations from–O=S=O– groups in SPEEK [80]. Table 1 shows the assignment of pointer colors to bands at different wavelengths. The region of 400–1800 cm^−1^ was deconvoluted in Figure 6 to determine if there were any band changes. A significant difference was at 1082 cm^−1^, associated with the PO_4_^3^^−^ group from the possible formation of some phosphate associated with the phosphorene addition. PO_4_^3^^−^ groups form absorption bands at 560–600 cm^−1^ and at 1000–1100 cm^−1^ [81].

### 3.3. TEM Analysis

To understand the effect of water on nanoparticle size, TEM images were obtained. From Figure 7, it is observed that 2D phosphorene formed distinct spherical nanoparticles in NMP; however, in water, the spherical nanoparticles agglomerated into clusters. pH has a large effect on nanoparticle agglomerate size [82]. Nanoparticle systems comprise of the nanoparticle and the suspension medium and the flux of hydrogen ions within the system controls agglomeration [82]. At pH levels close to the isoelectronic point of the nanoparticle, agglomeration is promoted. The isoelectronic point for phosphorene is 3. When phosphorene is suspended in NMP, the sodium hydroxide added during the exfoliation step led the system to become more basic, and hence the agglomeration effect was reduced; on the other hand, when the nanoparticles were rinsed in water, the system became more acidic because of the formation of some phosphoric acid and so agglomeration was favored. Observed nanoparticle size after exfoliation averaged about 5 ± 0.3 nm in NMP; however, if the solvent used were changed from NMP to water, the nanoparticle size was observed to increase to an average of 2 ± 0.9 μm. This occurred likely due to the tendency of the nanoparticles remain discrete in NMP, while agglomerating in water.

### 3.4. Morphologic Characterization of Membranes

Results from the optical profilometer indicated that the SPEEK:PSf membranes had smoother surfaces than those of the phosphorene membranes. Although the conventional method of quantifying roughness involves reporting the line roughness parameters, average line roughness (Ra), root mean square line roughness (Rq) and the mean depth of line roughness (Rz) [83], reporting the entire surface roughness parameters (Sa, Sq and Sz) [83] provides a greater understanding of the roughness over the entire surface measured than just the line measurement which could vary based on line location. From Figure 8, it was determined that the average surface roughness (Sa), root mean square roughness (Sq) and mean roughness depth (Sz) had values of 0.18, 0.24 and 4.95 μm, respectively. On the other hand, Figure 9 showed that the phosphorene membranes showed higher values of Sa = 0.45 μm, Sq = 0.61 μm and Sz = 6.46 μm. With phosphorene nanoparticles synthesized using the basic exfoliation technique agglomerates in water, agglomeration may be a cause of the observed increase in roughness of the phosphorene membranes since while phosphorene was added to the dope containing NMP, the non-solvent of the phase-inversion membrane casting process was water.

Previous studies also provided other morphologically significant parameters associated with the polymer evolution due to the addition of phosphorene [6]. The pore diameter at the maximum pore distribution that is, the most prevalent pore size of the SPEEK:PSf membranes was on average 0.022 μm (with smallest and largest detected pores being 0.017 and 0.086 μm), while that of the phosphorene membranes averaged 0.0024 μm (with smallest and largest detected pores being 0.0022 and 0.0078 μm). This further indicates the addition of phosphorene accumulating within the pores in agreement with agglomeration results. Furthermore, the contact angles of SPEEK:PSf and phosphorene membranes were found to be 48.3° ± 0.67° and 81.5° ± 0.64°, respectively and the increase in hydrophobicity was associated with the presence of the more hydrophobic phosphorene [84]. Lastly, it was observed that at a pH of approximately 6, the zeta potential of SPEEK:PSf was −61 ± 4.6 mV while that of the phosphorene membranes was −44 ± 7 mV, which was due to the phosphorene nanoparticles masking some of the sulfonic sites.

### 3.5. Phosphorene Leaching

The transport of two-dimensional phosphorene in porous media is largely dependent on the pH [85]. At a pH far away from the pH of the point of zero charge (pH_ZPC_), also known as the isoelectric point, the repulsive forces on the electric double layer on hydrated nanoparticles decrease, as a result there is lower aggregation and more dispersity [86]. The pH_ZPC_ of hydrated phosphorene is 3.0 [20]. Phosphorene nanoparticles were incorporated into membranes and their stability under acidic, basic, and neutral environments were determined using the ICP-OES. As seen in Figure 10, generally the nanoparticles seemed to be stable under all three conditions, with less than a 1% loss in amount of phosphorene. Under the basic medium though, irrespective of the initial concentration of phosphorene, the loss of phosphorene was highest, than other media. This could be as result of the pH of the basic media being too far from 3.0 thus leading to the lesser aggregation and more detectable free phosphorene. This factor also explains why as the concentration of phosphorene increased the basic dissolution increased.

### 3.6. Pore Structure Comparison

Pore size and porosity largely determine the efficiency of separation n [87], and these properties of the membrane are controlled by the fabrication technique [88]. These techniques include phase separation processes [89], stretching, track etching [90] and sintering [91] among others. For phase separation processes that use immersion precipitation like the nonsolvent-induced phase separation (NIPS) technique, liquid–liquid demixing controls the morphology of the membrane [87]. The structure of membranes obtained via phase immersion precipitation can be classified under five broad categories based on the polymer, solvent and non-solvent combination. They include noodles, cellular structures, macrovoids, bicontinuous structures and unconnected latex [92]. Membranes formed via the NIPS technique involving a solvent/nonsolvent combination of NMP and water, macrovoids (fingerlike in large quantities and pear shaped like in small quantities) are typically the kind of structures that represent a large portion of the membrane morphology [87,93]. Nodules, spherical beads which are fused together, are also typically observed on the surface layer, the layer where most separation occurs [94]. From the SEM images of both phosphorene and SPEEK:PSf membranes (Figure 11), nodules were observed on the surface scans and they gradually turned into macrovoids as expected based on the solvent/non-solvent combination used during the preparation of the membrane. Both membranes exhibited similar morphologic structures at the top and middle layers, but towards the bottom layer, there were noticeable differences, the SPEEK:PSf membranes merged into spherical macrovoids, while the phosphorene membranes retained its nodular/finger-like structures. This is similar to published studies using silver nanoparticles; thus, nanoparticles can act as pore formers increasing the length of the finger-like structures [65,66]. This was controlled by the low addition of phosphorene to the membranes, which again is based on observations from previous studies that use other nanoparticles.

### 3.7. Phosphorene Distribution on the Membrane

To ascertain the location of phosphorene nanoparticles within the membrane, a depth-profile scan was performed on the membrane. As seen in Figure 12, phosphorus was found to be present on the surface of the membrane and the amount increased as the etch-time increased, thus indicating that phosphorene was also present within the pores of the membranes. That is, even though there was some agglomeration of membranes, which was attributed to the increase in roughness observed, phosphorene was still found to be dispersed throughout the membrane matrix.

### 3.8. Flux Discussion

The phosphorene membranes showed a rejection of 78% ± 4% for BSA, while rejection for the SPEEK:PSf membranes was 43% ± 16%. This can be explained by the pore size of the membranes. Bovine serum albumin has a molecular weight of 66 kDa. The mean pore diameter of SPEEK:PSf and phosphorene membranes had been previously determined to be 0.022 and 0.0024 μm [6], respectively. With respect to permeability, Figure 13A shows that for SPEEK:PSf membranes, the average initial flux was 100 ± 12 LMH, final flux 23 ± 3 LMH and recovered flux after reverse flow filtration 65 ± 9 LMH. For phosphorene membranes (Figure 13B), the initial flux was 86 ± 40 LMH, final flux 8 ± 6 LMH and recovered flux after reverse flow filtration 30 ± 14 LMH. The smaller pores of the phosphorene membranes along with increased hydrophobicity may have led to observed rejection values along with reduced the flux values observed with phosphorene membranes. The low observed recovered flux after reverse-flow filtration also indicates an increase in the organic matter fouling layer on the phosphorene membrane, which agrees with the increased hydrophobicity of the membranes and decreased.

### 3.9. Operational Performance of Phosphorene Membranes

BSA filtration results show that while the rejection was increased, the flux during operation decreased, and more concerning, a fouling layer accumulated. BSA has an isoelectric point at pH 4.5–5.0 [95], so the protein is negatively charged at the neutral pH values of operation. BSA is also a large molecule of approximately 66.5 kDa. To study both the irreversibility of fouling and the potential of an ultraviolet (UV) light response by phosphorene membranes, the filtration of methylene blue (MB) was used so that the fouling layer could be visually observed. MB was chosen since it is known to degrade under UV light, is a hydrophobic and basic dye (MB+), and has an approximate 300 Daltons; therefore, while significantly smaller than BSA, it was expected to more irreversibly adsorb to membranes than BSA. Studies were performed using both visible and UV light sources to determine if an improvement was observed. Table 2 summarizes the flux values obtained, all performed at a constant pressure of 2.06 bar.

SPEEK:PSf membranes operated both under visible and UV light sources showed large declines in flux during MB filtration. Furthermore, cleaning using reverse flow filtration did not show any significant recovery in flux; therefore, membranes were irreversibly fouled. On the other hand, for phosphorene membranes, under visible light, the recovered flux after reverse-flow filtration to simulate cleaning with pure water was low, indicating an irreversible accumulation of MB on the membrane surface. This agrees with BSA filtration results that showed a significant decline in flux during BSA filtration along with a small flux recovery. However, when phosphorene membranes operated under a UV light source, a full recovery of flux after reverse flow filtration was observed. Membrane surfaces were then imaged and surface coverage, shown in Table 2, supports the removal of MB from the membrane surface. However, it was beyond the scope of this study to determine if the removal of the MB layer was physical (due to desorption) or chemical (due to degradation).

### 3.10. Toxicity of Phosphorene

The effect of a phosphorene exposure was tested on *C. elegans* mortality and reproduction in two different media, which differed by ionic strength and pH. The exposure in K medium with higher ionic strength and lower pH of 5.8 demonstrated lower toxicity than exposure in MHRW with higher pH of 7.8 and lower ionic strength. In fact, there was no mortality observed when the nematodes were exposed in K medium with phosphorous concentration up to 60 mg/L. From Figure 14, it can be seen that the exposure in MHRW resulted in concentration-dependent mortality with significant increase observed only at phosphorous concentration of 45 mg/L and above.

For reproduction, a low toxicity was documented in both media (Figure 15). Reproduction is more sensitive endpoint than mortality, and there were significant decreases in *C. elegans* reproduction at phosphorous concentrations at 12 mg/L in K medium and at 2.2 mg/L in MHRW.

These results demonstrate that toxicity of phosphorene in free form depends on the exposure media and it is critical to ensure that no leaching of phosphorene occurs when these nanoparticles are incorporated into a membrane. We have measured concentration of phosphorous in the permeates of the first, third and fifth order of filtration performed through the phosphorene membrane with K medium or MHRW. The levels of the phosphorous in the permeates were all below detection limit and there was no toxicity observed when the nematodes were exposed to the permeates. Thus, our results demonstrate that even though there was a reproductive toxicity observed in *C. elegans* exposed to free phosphorene, the release of the phosphorene from the membrane was minimal or none and did not cause toxicity. This is a step in the right direction of developing safe polymeric phosphorene membranes that can be eventually applied in removal of organic pollutants. Since the relatively low toxicity was observed for the phosphorene exposures, it is imperative to examine effect of phosphorene on toxicity under different conditions as well as effects of phosphorene on other sublethal endpoints. For instance, our results above showed that less than 1% release of phosphorene from the membranes can occur at basic conditions, and thus further studies are warranted to examine additional factors that may promote phosphorene release.

## 4. Conclusions

Phosphorene membranes were synthesized to further characterize the evolution of the polymeric membrane fabrication upon addition to phosphorene. It was observed that phosphorene formed spherical distinct nanoparticles after exfoliation in basic-NMP and clustered spherical nanoparticles in water because of the effect of the flux of hydrogen ions (H^+^) within the nanoparticle system. In leaching studies, it was observed that phosphorene loss was less than 1% of the initial amount of phosphorene added, implying stability within the membrane matrix. Depth profile scans of phosphorene membranes showed that phosphorene nanoparticles were dispersed both on the surface of the membrane and within the pores of the membrane, indicating that while agglomeration may have occurred, phosphorene was still dispersed throughout the membrane matrix. Surface morphology studies indicated that phosphorene membranes had rougher surfaces, while the SPEEK:PSf membranes had smoother surfaces, which was likely due to some agglomeration caused by water being used as the nonsolvent during membrane fabrication via NIPS. The membranes modified with phosphorene displayed a higher protein rejection, but lower flux values and flux recovery after filtration possibly due to the decrease in average pore size. Toxicity results show that exposure to a phosphorene in a free form caused a relatively low toxicity in *C. elegans* with reproduction being a more sensitive endpoint than mortality. In addition, toxicity differed when exposures were conducted in two different media with MHRW showing higher toxicity. However, permeates of the same media through phosphorene membrane did not show toxicity due to minimal release of the phosphorene from the membranes further buttressing that phosphorene remained bound during filtration. Thus, phosphorene-based membranes open a new field for research in membrane science since phosphorene nanoparticles synthesized were found to be stable within the membrane structure, with less than 1% leaching of phosphorene. The toxicity of the free phosphorene indicate that it is critical to continue studies examining fate of phosphorene incorporated into the membranes under different environmental conditions in order to develop safe phosphorene membranes.

## Figures and Tables

**Figure 1 polymers-12-01555-f001:**
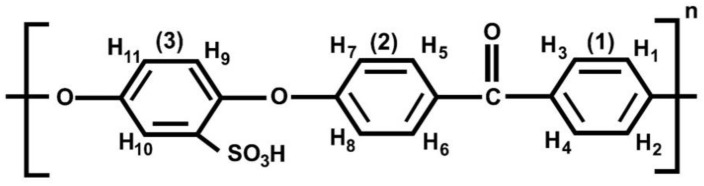
Nomenclature of aromatic protons of sulfonated poly ether ether ketone (SPEEK).

**Figure 2 polymers-12-01555-f002:**
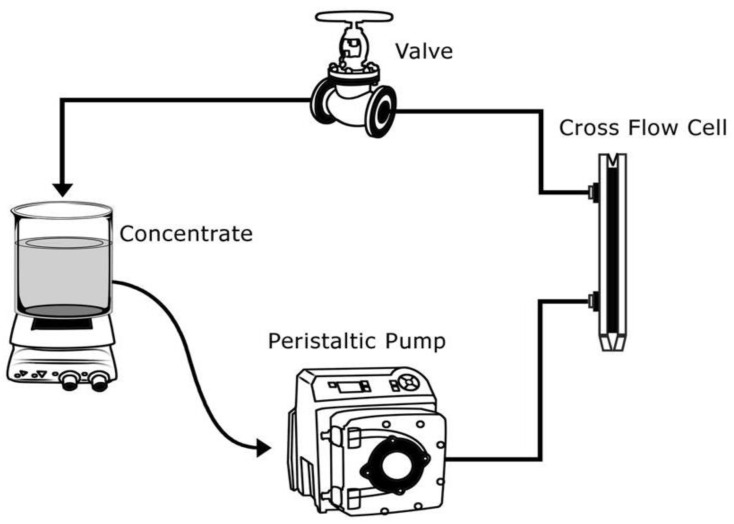
Schematic of the leaching study setup.

**Figure 3 polymers-12-01555-f003:**
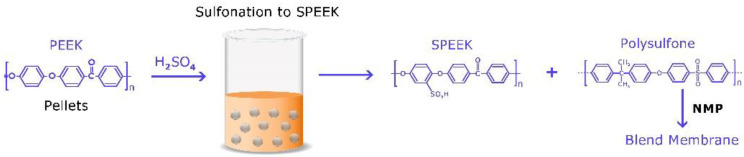
Fabrication of the blended polymeric membrane.

**Figure 4 polymers-12-01555-f004:**
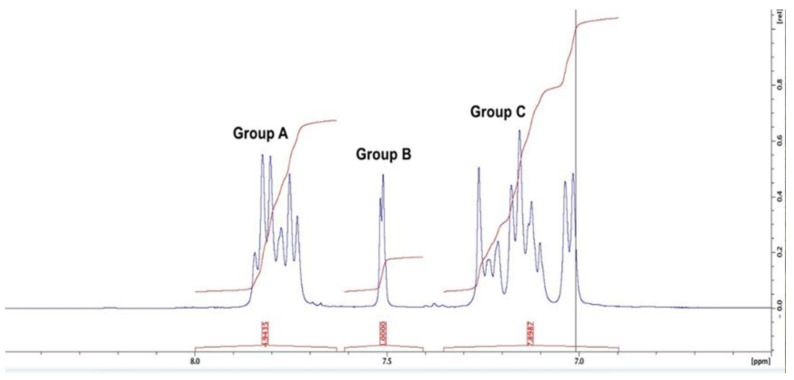
^1^H NMR spectrum of SPEEK in dimethyl sulfoxide.

**Figure 5 polymers-12-01555-f005:**
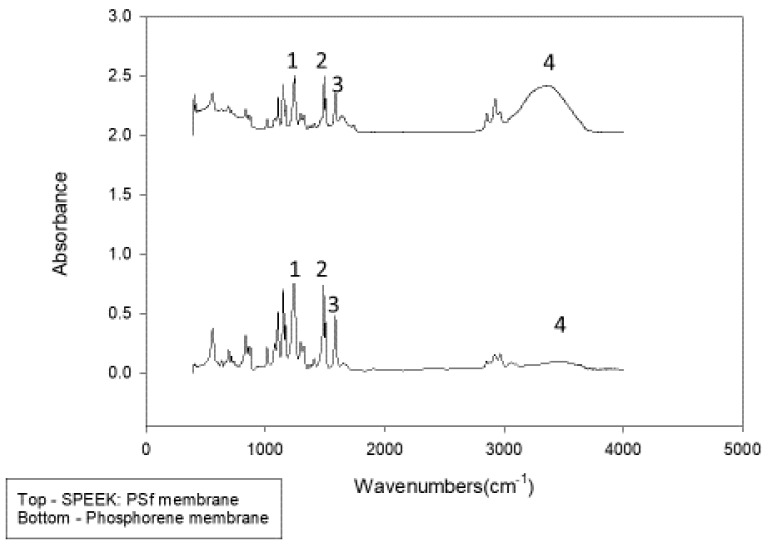
FTIR spectra of phosphorene and SPEEK: Polysulfone (PSf) membranes over the region 400–4000 cm^−^^1.^

**Figure 6 polymers-12-01555-f006:**
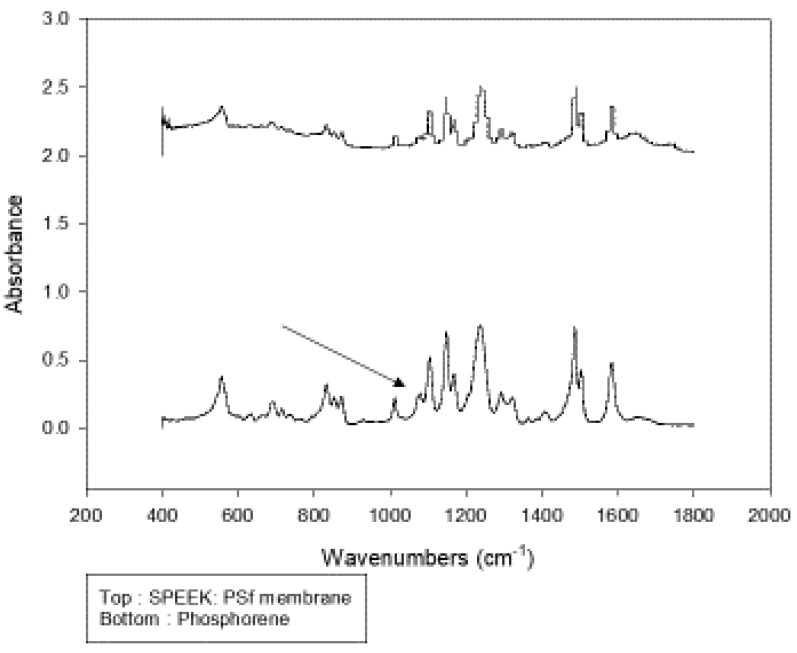
FTIR spectra of phosphorene and SPEEK:PSf membranes over the region 400–1800 cm^−1^.

**Figure 7 polymers-12-01555-f007:**
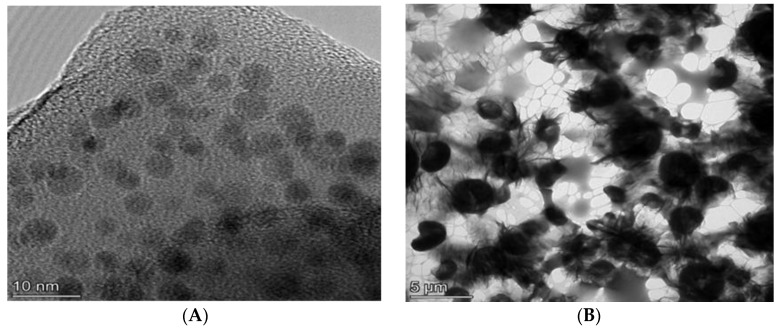
TEM images of 2D phosphorene in different solvents. (**A**) *N*-methyl pyrrolidone (NMP); (**B**) water.

**Figure 8 polymers-12-01555-f008:**
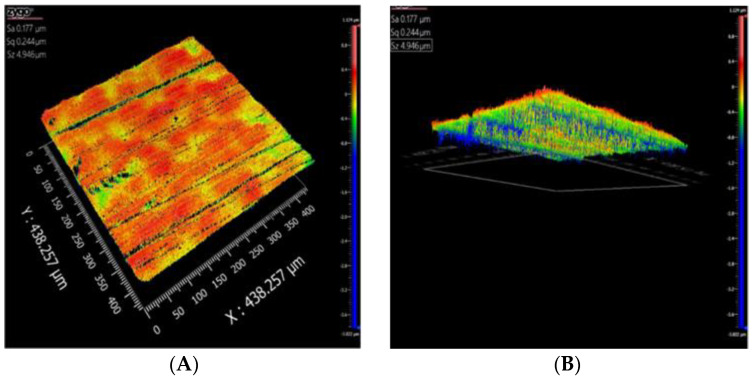
(**A**) Three-dimensional- and (**B**) bottom-view of SPEEK:PSf membranes showing the surface morphology.

**Figure 9 polymers-12-01555-f009:**
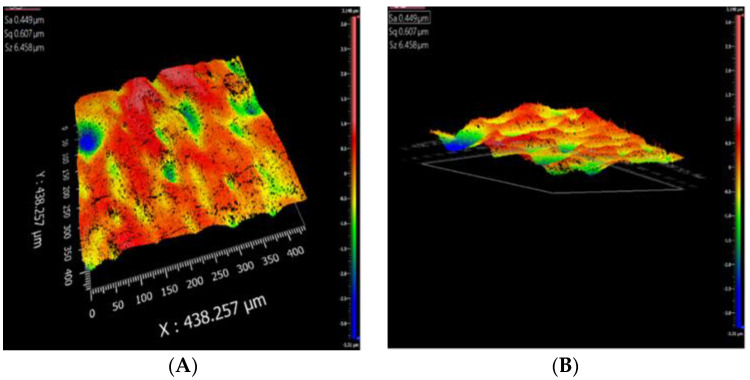
(**A**) Three-dimensional- and (**B**) bottom-view of phosphorene membranes showing the surface morphology.

**Figure 10 polymers-12-01555-f010:**
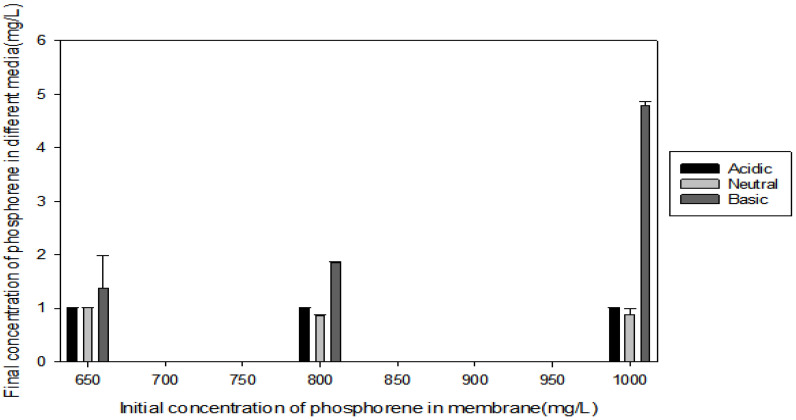
Leaching experiment of phosphorene in membranes.

**Figure 11 polymers-12-01555-f011:**
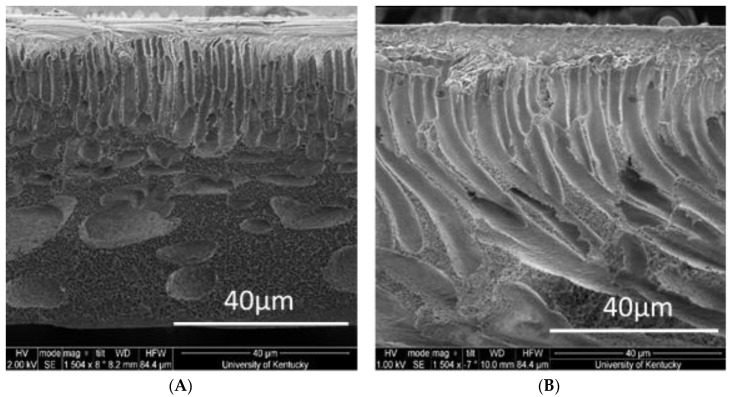
Cross-section images of (**A**) SPEEK:PSf membrane and (**B**) and phosphorene membrane.

**Figure 12 polymers-12-01555-f012:**
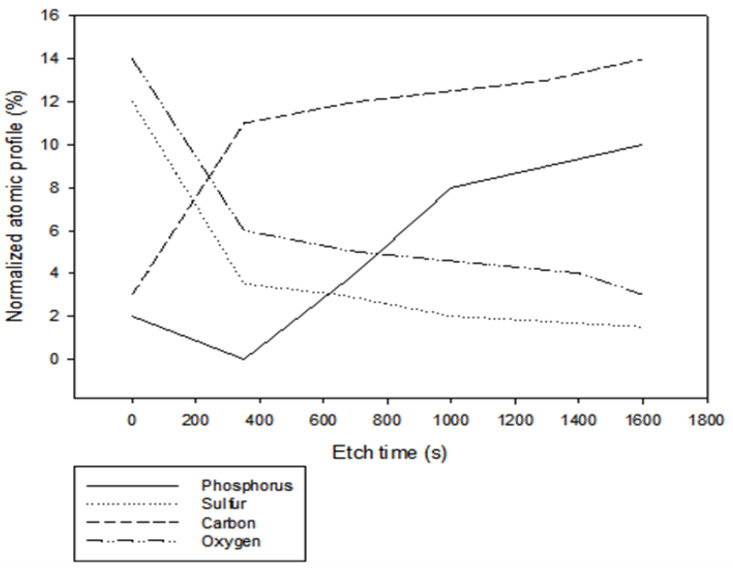
Depth-profile scan on phosphorene membranes.

**Figure 13 polymers-12-01555-f013:**
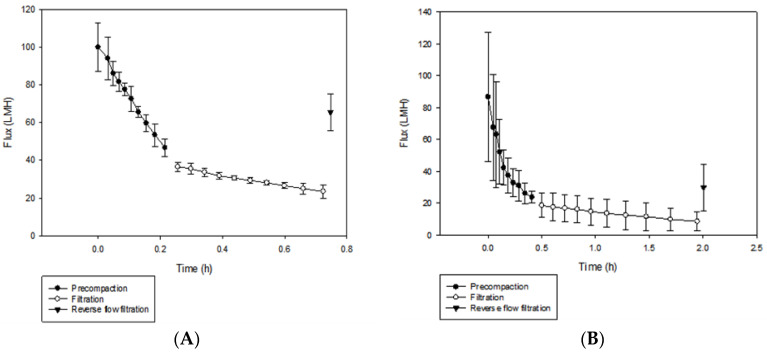
(**A**) Flux analysis of SPEEK:PSf membranes and (**B**) phosphorene membranes.

**Figure 14 polymers-12-01555-f014:**
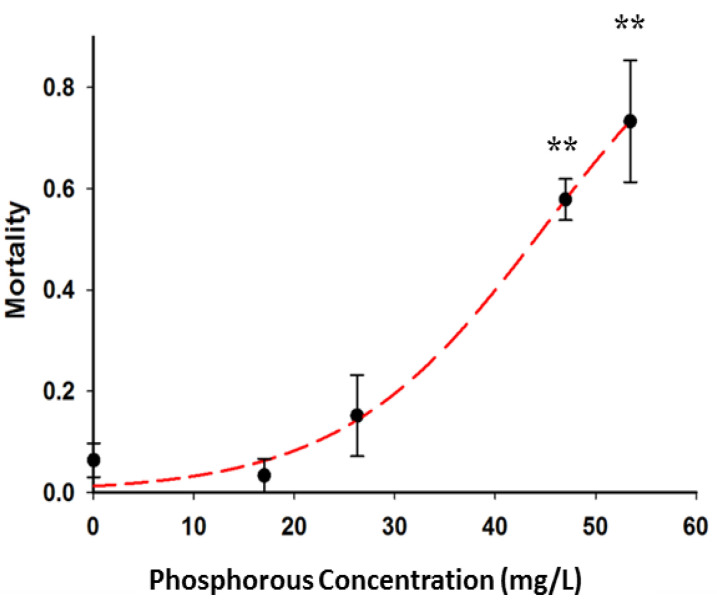
*Caenorhabditis elegans* mortality in response to phosphorene exposure over 24 h in moderately hard reconstituted water (MHRW).The double asterisks indicates statistical significance at *p* < 0.01.

**Figure 15 polymers-12-01555-f015:**
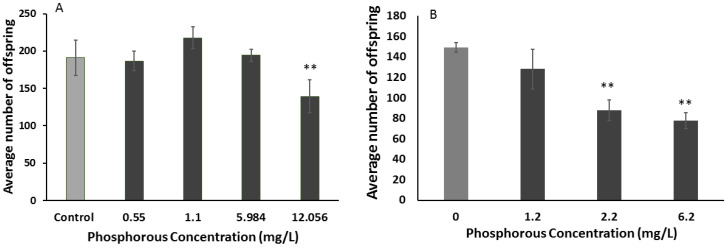
Effect of free phosphorene exposure over 24 h on reproduction in *Caenorhabditis elegans* in (**A**) K medium and (**B**) moderately hard reconstituted water (MHRW). The double asterisks indicates statistical significance at *p* < 0.01.

**Table 1 polymers-12-01555-t001:** Assignment of FTIR bands at different wavelengths.

Number	Wavelength Number (cm^−1^)	Functional Group
1	1230	–O=S=O–
2	1487	C=C
3	1586	C=C
4	3450	OH

**Table 2 polymers-12-01555-t002:** Flux values of phosphorene membranes operated under visible and UV light sources.

	SPEEK:PSf Membranes		Phosphorene Membranes	
Flux (LMH)	Visible	UV	Visible	UV
PWF initial	126	67	56	107
PWF final	92	37	74	82
MB initial	72	42	71	68.1
MB final	43	29	42	31
Recovered	40	17	25	70
Normalized membrane surface coverage (%)	100	95	76	30

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
