# Peer review of "Nanohybrid Membrane Synthesis with Phosphorene Nanoparticles: A Study of the Addition, Stability and Toxicity"

_polymers, 2020, doi:10.3390/polym12071555_

Round 1

Reviewer 1 Report

It is a good study with overall adequate presentation. Some comments are given below:

1) The novelty of this work must be further highlighted in the Introduction section, emphasizing which is the major difference with other similar works.

2) The Discussion must be drastically improved. The whole manuscript has 83 Refs, however the Results and Discussion section has 13 Refs from 83! This not acceptable. Please try to further state the discussion part given comparison with literature. 

3) Some sentences must be revised for typos, grammar and syntax errors. There are many parts in the whole manuscript.

4) The References must be more updated.

Author Response

We thank the editor and the reviewers for all recommended improvements. To address the comments and make it clear to follow, we use the BOLD format for our reply to the reviewers’ comments, and then use the format of RED and BOLD to make the changes

Reviewer 1

1) The novelty of this work must be further highlighted in the Introduction section, emphasizing which is the major difference with other similar works.

The following was added to the Introduction Section of the manuscript:

As more researchers turn to two dimensional materials for membrane modifications, the need for a 2D material that inherently allows fine tuning towards membrane enhancement is pertinent. Graphene has no band gap and other 2D transitional metal dichalcogenides (TMDs) possess band gaps only as monolayers[1]. Phosphorene has direct band gaps in all its three forms, bulk, monolayer and few layers[1]. Phosphorene has also been studied for its electrocatalytic properties, which research shows outperforms ruthenium (iv) oxide and Co3O4 /N-graphene[2].  Currently, while a large bulk of experimental research efforts has focused towards producing air stable phosphorene[1-4], there is limited information on the incorporation of phosphorene in membranes as well as a thorough understanding of its physicochemical properties when utilized as a membrane additive. In a previous study[5], the photocatalytic properties of phosphorene-based membranes were examined; on the other hand, in this study, the effects of phosphorene on the morphological structure of the polymeric blend along with the evolution of the modifications were investigated. Furthermore, we discuss its stability under several pH environments as well as study biological effects of phosphorene-based membrane permeates on a nematode.

2) The Discussion must be drastically improved. The whole manuscript has 83 Refs; however, the Results and Discussion section has 13 Refs from 83! This not acceptable. Please try to further state the discussion part given comparison with literature

We agree with the reviewer that an extensive discussion from literature is important and we did so while preparing this manuscript. The discussion part of the paper does contain references from literature, but please note that this research is novel and currently to the best of our knowledge, no other group is researching on phosphorene in membranes. The following references were added to the discussion section of the manuscript:

Found under the subsection Pore Structure Comparison: These techniques include phase separation processes[6], stretching, track etching[7] and sintering[8] amongst others,

Found under the subsection Operational Performance of Phosphorene Membranes: BSA has an isoelectric point at pH 4.5–5.0 [9], so the protein is negatively charged at the neutral pH values of operation

3) Some sentences must be revised for typos, grammar, and syntax errors. There are many parts in the whole manuscript

The reviewer is right, a badly written manuscript will hinder the assimilation of the content of the manuscript. The three native-English speaking authors have proofread the manuscript carefully.

4) The References must be more updated.

While answering the previous comments, more recent papers were cited and, the following has been added to the introduction section of the manuscript

The main issue with phosphorene is the fast degradation under ambient conditions as a result of the generation of reactive oxygen species. Research however has shown that incorporating phosphorene into polymers preserves the structure and properties of phosphorene[10, 11]

The following was also added to the manuscript

GPW was supported by Summer Undergraduate Research in Environmental Sciences (SURES) funded by the National Institute of Environmental Health Sciences (NIEHS) R25ES027684.

The title was modified to: Nanohybrid Membrane Synthesis with Phosphorene Nanoparticles: A Study of the Addition, Stability and Toxicity

References:

  1. Sang, D.K., H. Wang, Z. Guo, N. Xie, and H. Zhang, Recent Developments in Stability and Passivation Techniques of Phosphorene toward NextGeneration Device Applications. Advanced Functional Materials, 2019. 29(45): p. 1903419.
  2. Dinh, K.N., Y. Zhang, J. Zhu, and W. Sun, Phosphorenebased Electrocatalysts. Chemistry–A European Journal, 2020.
  3. Qu, Z., K. Wu, W. Meng, B. Nan, Z. Hu, C.-a. Xu, Z. Tan, Q. Zhang, H. Meng, and J. Shi, Surface Coordination of Black Phosphorene for Excellent Stability, Flame Retardancy and Thermal Conductivity in Epoxy Resin. Chemical Engineering Journal, 2020: p. 125416.
  4. Li, H., P. Lian, Q. Lu, J. Chen, R. Hou, and Y. Mei, Excellent air and water stability of two-dimensional black phosphorene/MXene heterostructure. Materials Research Express, 2019. 6(6): p. 065504.
  5. Eke, J., K. Elder, and I.C. Escobar, Self-cleaning nanocomposite membranes with phosphorene-based pore fillers for water treatment. Membranes, 2018. 8(3): p. 79.
  6. Pagliero, M., A. Bottino, A. Comite, and C. Costa, Novel hydrophobic PVDF membranes prepared by nonsolvent induced phase separation for membrane distillation. Journal of Membrane Science, 2020. 596: p. 117575.
  7. Kaya, D. and K. Keçeci, Track-Etched Nanoporous Polymer Membranes as Sensors: A Review. Journal of The Electrochemical Society, 2020. 167(3): p. 037543.
  8. Yu, L., M. Kanezashi, H. Nagasawa, and T. Tsuru, Phase inversion/sintering-induced porous ceramic microsheet membranes for high-quality separation of oily wastewater. Journal of Membrane Science, 2020. 595: p. 117477.
  9. Sah, B.K., K. Das, and S. Kundu, pH-dependent structure, pattern and hysteresis behaviour of lipid (DMPA)-protein (BSA) monolayer complex. Colloids and Surfaces A: Physicochemical and Engineering Aspects, 2019. 579: p. 123663.
  10. Fonsaca, J.E., S.H. Domingues, E.S. Orth, and A.J. Zarbin, Air stable black phosphorous in polyaniline-based nanocomposite. Scientific reports, 2017. 7(1): p. 1-9.
  11. Peruzzini, M., R. Bini, M. Bolognesi, M. Caporali, M. Ceppatelli, F. Cicogna, S. Coiai, S. Heun, A. Ienco, and I.I. Benito, A perspective on recent advances in Phosphorene functionalization and its applications in devices. European journal of inorganic chemistry, 2019. 2019(11-12): p. 1476-1494.

Reviewer 2 Report

This is an interesting study, however a few scientific and technical issues need to be addressed before it is considered for publication by Polymers.

  1. It is not clear what is the real content of phosphorene in the membrane. The authors only said "The dope solution consisted of a (95/5%) ratio of PSf and SPEEK, and 0.5 wt.% of exfoliated phosphorene in NMP.". In general, the solvent NMP was fully eliminated from the final membrane product. Therefore, "0.5wt% in NMP" can not be used as an indicator of phosphorene in the membrane. It is critical for authors to provide detail experimental information on membrane contents, rather than the drop solution.
  2. Any control experiments on various phosphorene concentration, except the"0.5wt% in NMP"?
  3. In Figure 11, the SEM images of cross-section shows clear difference between the samples with and without phosphorene. However, the authors claimed that "Phosphorene itself was in low enough concentrations that it did not pose any significant effects on the morphology of the membranes.". This claim is not right. A proper revision and analysis (rewrite this section) is needed.

Author Response

We thank the editor and the reviewers for all recommended improvements. To address the comments and make it clear to follow, we use the BOLD format for our reply to the reviewers’ comments, and then use the format of RED and BOLD to make the changes

Reviewer 2:

It is not clear what is the real content of phosphorene in the membrane. The authors only said, "The dope solution consisted of a (95/5%) ratio of PSf and SPEEK, and 0.5 wt.% of exfoliated phosphorene in NMP.". In general, the solvent NMP was fully eliminated from the final membrane product. Therefore, "0.5wt% in NMP" cannot be used as an indicator of phosphorene in the membrane. It is critical for authors to provide detail experimental information on membrane contents, rather than the drop solution.

The reviewer is right; the right amount of phosphorene should be stated for replication studies. Phosphorene does not dissolve in NMP, but polysulfone and sulfonated poly ether ether ketone dissolve in NMP.  Yes, the solvent was mostly eliminated during the phase inversion process, and what we had left behind was phosphorene and the polymer blend. There may have been some loss of phosphorene during the casting process, but it was not noticeable. We showed that there was negligible loss of phosphorene from the membrane via the leaching experiment, so we can assume that almost all the phosphorene remained bound within the membrane, this we also confirmed with the atomic profile study, as shown in Section 3.7 and Figure 12. We also carried out an FTIR study to understand the composition of the membrane. As shown in Figure 12, the normalized atomic profile percentage of phosphorus varied depending on location (i.e., by layer); thus, it would be difficult measure the exact amount of phosphorene in the membrane. The following was added to the method section of the manuscript in section 2.12:

During the phase inversion process, some loss of phosphorene may have occurred, but this was unnoticeable. The remnant coagulant bath solution was tested after casting for phosphorus which was below the detection limit of 50 ng/mL. 0.5% w/v of phosphorene was used (5 mg/mL) during the fabrication of the membrane. Since no loss was detected, therefore, the theoretical percentage of phosphorene in the membrane was 0.5% w/v.

Any control experiments on various phosphorene concentration, except the"0.5wt% in NMP"?

Yes, we experimented with other concentrations of phosphorene, but this was our most successful attempt. Since nanoparticles can change the morphological structure of membranes by acting as pore formers, keeping them at a low concentration helps balance the tradeoff of their positive impact on their negative impacts. For the purpose of this study, the amount of phosphorene was constant in all the membranes. This paper describes the modification evolution and detailed results based on the success of our initial published short communication entitled “Self-cleaning nanocomposite membranes with phosphorene-based pore fillers for water treatment." Membranes 8.3 (2018): 79. Therefore, for consistency, we used the same amount of phosphorene in the membranes here. The following was added to Section 2.12 to clarify

Since nanoparticles can change the morphological structure of membranes by acting as pore formers, keeping the concentration to a low 0.5 wt% helped balance the tradeoff of their positive impact on their negative impacts on the membrane [1, 2].  

In Figure 11, the SEM images of cross-section show clear difference between the samples with and without phosphorene. However, the authors claimed that "Phosphorene itself was in low enough concentrations that it did not pose any significant effects on the morphology of the membranes.". This claim is not right. A proper revision and analysis (rewrite this section) is needed

The reviewer is correct. From literature[3], we know that membranes formed via the NIPS technique with NMP and water usually display finger-like structures at the topmost layer where separations occurs. From the SEM images, we observe these structures occurring in both membranes; hence, the discussion that there was no significant structural change on the morphology of the membranes; however, that should have been clarified that this observation was closer to the active layer of the membranes. The reviewer is right in noticing that for the SPEEK: PSf membrane, these finger-like structures end early and spherical macrovoids form. On the other hand, phosphorene membranes have the finger-like structures going farther through to the length of the membranes. We have previously observed this in other work with nanoparticles; that is, nanoparticles can act as pore formers increasing the length of the finger-like structures[1, 2]. This was controlled by the low addition of phosphorene to the membranes, which again is based on observations from previous studies that use other nanoparticles. Therefore, the following was added to section 3.6 of the manuscript:

Both membranes exhibited similar morphological structures at the top and middle layers, but towards the bottom layer, there were noticeable differences, the SPEEK: PSf membranes merged into spherical macrovoids, while the phosphorene membranes retained its nodular/finger-like structures. This is similar to published studies using silver nanoparticles; that is, nanoparticles can act as pore formers increasing the length of the finger-like structures[1, 2]. This was controlled by the low addition of phosphorene to the membranes, which again is based on observations from previous studies that use other nanoparticles.

References:

  1. Sprick, C., S. Chede, V. Oyanedel-Craver, and I.C. Escobar, Bio-inspired immobilization of casein-coated silver nanoparticles on cellulose acetate membranes for biofouling control. Journal of Environmental Chemical Engineering, 2018. 6(2): p. 2480-2491.
  2. Dong, X., H.D. Shannon, A. Amirsoleimani, G.M. Brion, and I.C. Escobar, Thiol-Affinity Immobilization of Casein-Coated Silver Nanoparticles on Polymeric Membranes for Biofouling Control. Polymers, 2019. 11(12): p. 2057.
  3. Frommer, M.A. and D. Lancet, The mechanism of membrane formation: membrane structures and their relation to preparation conditions, in Reverse Osmosis Membrane Research. 1972, Springer. p. 85-110.

Round 2

Reviewer 1 Report

All my comments of the initial submission have been correctly replied and included in the revised manuscript. The quality of this work has been drastically improved after revision and therefore I recommend its publication as it is.

Reviewer 2 Report

The authors have fully addressed reviewers' comments. Therefore, i recommend it ti be accepted at its current stage.